# Multiple Machine Learning Approaches for Morphometric Parameters in Prediction of Hydrocephalus

**DOI:** 10.3390/brainsci12111484

**Published:** 2022-11-01

**Authors:** Hao Xu, Xiang Fang, Xiaolei Jing, Dejun Bao, Chaoshi Niu

**Affiliations:** 1Department of Neurosurgery, The First Affiliated Hospital of USTC, Division of Life Sciences and Medicine, University of Science and Technology of China, Hefei 230001, China; 2Department of Neurology, The First Affiliated Hospital of USTC, Division of Life Sciences and Medicine, University of Science and Technology of China, Hefei 230001, China

**Keywords:** hydrocephalus, imaging diagnosis, machine learning

## Abstract

Background: The diagnosis of hydrocephalus is mainly based on imaging findings. However, the significance of many imaging indicators may change, especially in some degenerative diseases, and even lead to misdiagnosis. Methods: This study explored the effectiveness of commonly used morphological parameters and typical radiographic findings in hydrocephalus diagnosis. The patients’ imaging data were divided into three groups, including the hydrocephalus group, the symptomatic group, and the normal control group. The diagnostic validity and weight of various parameters were compared between groups by multiple machine learning methods. Results: Our results demonstrated that Evans’ ratio is the most valuable diagnostic indicator compared to the hydrocephalus group and the normal control group. But frontal horns’ ratio is more useful in diagnosing patients with symptoms. Meanwhile, the sign of disproportionately enlarged subarachnoid space and third ventricle enlargement could be effective diagnostic indicators in all situations. Conclusion: Both morphometric parameters and radiological features were essential in diagnosing hydrocephalus, but the weights are different in different situations. The machine learning approaches can be applied to optimize the diagnosis of other diseases and consistently update the clinical diagnostic criteria.

## 1. Introduction

Hydrocephalus is caused by an imbalance of production and absorption of cerebrospinal fluid (CSF) or obstruction of its pathways, resulting in ventricular dilatation and increased intracranial pressure. Imaging finding plays a crucial role in the diagnosis. The sign of hydrocephalus is ventricular enlargement with some typical radiographic features, including dilated temporal horns, dilated third ventricle, rounded poster horns, thinned corpus callosum, and CSF extravasation [1,2]. However, ventricles of the human brain also enlarged with aging, neurodegenerative diseases, intrinsic and extrinsic pathologies [3,4]. The classical hydrocephalus diagnostic indicators have been widely used for a long time. However, the value and weight of each diagnostic indicator has not been studied.

The morphometric examination of neuroimages is a practical approach to assess structural changes of hydrocephalus [5,6,7]. In this study, the commonly used morphological parameters, including frontal horns’ length (FHL), maximum lateral length (MLL), biparietal diameter (BPD), Evans’ ratio (ER), cella media ratio (CMR), and frontal horns’ ratio (FHR), were collected and compared. The classic radiographic features were also analyzed, including dilated temporal horns, dilated third ventricle, rounded posterior horns, and disproportionately enlarged subarachnoid space hydrocephalus. Multiple machine learning approaches have been used to predict hydrocephalus and measure the weight of each diagnosis index. We believe that better prediction can significantly improve the diagnosis and treatment of these patients.

## 2. Materials and Methods

### 2.1. Patient Population

All patients attended our center between the years February 2017 and October 2021. The Inclusion criteria of this study is: The inclusion criteria: 1. Older than 14 years; 2. Ventricular enlargement in CT scan; 3. Consistent with symptoms related to hydrocephalus: urinary incontinence, ataxia, slow, progressive autonomous language body activity decreases, abnormal gait. Exclusion criteria: 1. Congenital hydrocephalus; 2. Malignant tumor and other serious systemic diseases; 3. Obstructive hydrocephalus; 4. Acute hydrocephalus. ALL the suspected patient had been carefully evaluated in our department, and one of the most important criteria is Tap test.

We had previously obtained baseline data for walking ability and psychometric tests on two separate occasions at the same time of day, on the first and second days after admission. After CSF removal and a rest period of one to three hours, these tests were repeated for comparison with the baseline data. The tests were done at the same time of day as the baseline tests. The test battery included the following: Walking test, reaction time, memory test and identical forms. The results obtained after CSF removal were compared with the best of the results from the two baseline tests. Criteria for clinical diagnosis of hydrocephalus is CSF tap-positive: 1. Criteria were a 10% or more improvement on the 10 MWT; 2. Improvement of 10% or more on the TUG; 3. An improvement of 3 points or more on MMSE; 4. The patient felt subjective relief of symptoms. The tap test was considered positive if two or more of the four different items improved after CSF removal [8,9,10,11].

298 admitted patients were included in the study according to the related symptoms of hydrocephalus, including urinary incontinence, ataxia, slow, progressive autonomous language body activity decreases. All patients were conducted CT scan pre and postoperative. 62 cases of them were finally diagnosed with hydrocephalus by lumbar puncture test confirmed. The diagnostic criteria for hydrocephalus refer to the latest diagnostic and treatment standards [12,13]. The VP shunts were implanted, and all patients were achieved positive outcomes in the at least 6 months follow up. The criteria of good outcomes are remission or disappearance of symptoms. The other 36 cases were not diagnosed hydrocephalus because lumbar tap test results did not support it. Most of them were finally diagnosed with Alzheimer’s or vascular dementia. All examinations and diagnoses were performed by two experienced neurosurgeons independently. At the same time, 200 normal patients were included as the control group. The group were negative individuals on routine head CT examination.

### 2.2. Measurements of Linear Parameters

All images were taken by a trained and experienced neuroradiologist and clinician in standardized condition and manner. Although the diagnostic value of MRI is irreplaceable, CT images, the preferred measurement of the parameters in practice, are more popular and convenient in the clinic. All patients had examination of the brain using the department’s Siemens 298 slice Somatom definition AS multi-detector spiral CT. Based on the measurements following linear indices were calculated [14,15]. All the measures used in the calculation of parameters are demonstrated in Table 1 and Figure 1 [16,17].

### 2.3. Model Construction

All features were selected by clinicians based on their experience in diagnosis before automatic analysis. Four typical machine learning algorithms were chosen to construct the prediction model, which were random forest (RF), support vector machine (SVM), artificial neural network (ANN), and extreme gradient boosting (XGBoost) [18,19,20]. A SVM tries to find optimal decision boundary—hyperplanes that best separate data of different categories. An ANN is a collection of connected nodes (neurons) that compute the output by some nonlinear functions of the sum of its input. During the training process, the connections (weights) between neurons are modified so that computers can learn the pattern to classify data. XGBoost is an ensemble learning method that constructs multiple decision trees to organize data [21,22].

The whole data samples were randomly split into training and test sets according to a division of 7:3. Optimal features and hyperparameters combinations for the model were determined on the training set. Fivefold cross-validation was used in the process of feature selection and hyperparameters(Figure 2). The important indicators of the machine learning model include precision and recall. Precision refers to the actual positive samples among all predicted positive samples. The formula is as follows: Precision = TP/(FP + TP). Recall refers to the probability of being predicted to be a positive sample in all samples. Its formula is as follows: Recall rate =TP/(TP + FN). To consider the two factors, F1 score were calculated as F1 = 2 precision-recall rate/(precision + recall rate). To assess the discriminative performance of this risk score in both the development and validation subsamples, we used the c-statistic, which was also displayed graphically as the area under the receiver operating characteristic (ROC) curve. An area under the ROC curve (AUC) of 0.5 indicates no discrimination, whereas an AUC of 1.0 indicates perfect discrimination.

## 3. Results

A total of 298 patients were included in the study. The baseline statistics of the hydrocephalus group, symptomatic group, and normal control group are presented in Table 2. The hydrocephalus group and symptomatic group were compared, and the ROC curves of the four derived models are plotted in Figure 3. The ANN model (0.96 ± 0.05) and RF (0.96 ± 0.06) achieved the highest area under the ROC curve, followed by the SVM (0.94 ± 0.05) model and xgBoost (0.94 ± 0.07). In the observation of the precision, recall, and F1 value, the SVM model performs relatively well, especially the F1 value reached 0.96 in the test set (Table 3).In terms of the weight of diagnostic features, the top three are DESH (28.11%), ER (24.79%), and FHR (10.64%).

Although the results were consistent with our clinical experience, the most important index requires clinician supervisor judgment. To further clarify the value of objective indicators, we removed radiological features, and the analysis was redone. In terms of morphometric parameters features, the ROC curve was not as superior as before but can still be valuable in the diagnosis of hydrocephalus. At this time, the SVM model achieved the highest area under the ROC curve of 0.74 ± 0.15, followed by the RF (0.73 ± 0.17) model, ANN (0.71 ± 0.16) model, and xgBoost (0.70 ± 0.18).

Then we attained radiographic features and confusion matrix of morphometric parameters for analysis. From the results, it is seen that the combined four radiographic features, including dilated temporal horns, dilated third ventricle, rounded posterior horns, and DESH, is leading in decision making, even without measuring parameters indicators (AUC = 0.94 ± 0.03) (Figure 4). In terms of morphometric parameters, ER, FHR and CMR showed similar diagnostic values (Table 4).

The following is the key part of this study. We analyze the patient with hydrocephalus symptoms. The ROC curves showed a visible decline compared to the previous comparison. The SVM (0.83 ± 0.06) model achieved the highest area, followed by the RF (0.78 ± 0.07) model, ANN (0.80 ± 0.09) model, and xgBoost (0.83 ± 0.06). The importance of features were DESH (46.88%), FML (15.27%), ER (12.25%), FHR (8.31%) and CMR (6.68%).The AUC and F1 were also inferior in terms of morphometric parameters features (Table 5). In terms of morphometric parameters, the AUC also declined (Figure 5). It is worth noting that the weight of ER decreased from 52.26% to 18.13%, while the weight of FHL (30.34%) and FHR (24.47%) were increased.

To further clarify the diagnostic value of these indicators, we performed regression analysis. The results show that ER, CMR, FHR, and DESH are significant (*p* < 0.05) in this diagnosis. Other diagnostic indicators showed no statistical significance. Then, we noticed that the patient in the symptomatic group (49.87 ± 15.53), who had symptoms but without hydrocephalus diagnosis, were relatively older than the hydrocephalus group (70.37 ± 11.42). Thus, we extract the patients aged more than 60 years and conduct another analysis (Table 6). Univariate analysis showed that there were statistically significant differences in ER, CMR, FHR, dilated third ventricle and DESH (*p* < 0.05). The dilated temporal horns and rounded posterior horns were not significant (*p* > 0.05). Meanwhile, aging may tend to be a negative diagnostic index of hydrocephalus (*p* = 0.002, OR = 1.14, 95% CI: 1.031~1.271).

## 4. Discussion

Hydrocephalus is a common symptom that can have a number of causes [1,23]. Hydrocephalus on its own can be life-threatening and very difficult to manage with many complications [7,24,25]. If the symptom is not treated, it can develop into an independent disease that requires ongoing treatment even after the causes are relieved. Even so, some of the patients may not be easy to be diagnosed and treated in time.

The diagnosis of hydrocephalus depends on clinical manifestations, imaging findings, and lumbar puncture results. Imaging plays a central role in confirming the diagnosis, identifying the cause, and planning treatment. As an invasive operation, lumbar puncture cannot be applied to the symptomatic patients as a general diagnostic method, so many patients would be undiagnosed and lose the best opportunity for treatment. Especially for a special form of hydrocephalus is known as “idiopathic normal pressure hydrocephalus,” imaging finding is irreplaceable [26,27,28,29].

A very sensitive sign of this is the dilatation of the temporal horns and posterior horns. A diameter of >2 mm in adults is considered pathological, but there are no strict criteria in the literature because of different head circumferences [2]. Moreover, the width of the third ventricle increases so that it is no longer slit-shaped but rather ballooned or laterally bowed. Compared to the dilated ventricular system, the external CSF spaces are disproportionately thin [30]. Because the imaging sign relies on clinical experience, many objective measures still play an important role in the diagnosis of hydrocephalus. The Evans’ Index is most widely used in the clinical routine to quantify dilatation of the ventricles. The ratio of the maximum width of the frontal horns of the lateral ventricles And the greatest internal diameter of the skull. It was described by Evans [31] in 1942 as a method of measurement of ventricular size in pediatric patients. A value of >0.3 is considered pathological [12,32]. Cella Media Ratio and Frontal Horns’ Ratio are also well-used indexes. Cella Media Ratio is the ratio of the minimum distance between lateral walls of lateral ventricles in cella media region, a, and maximum transverse (external) diameter. It is expected to be smaller than 0.25 in normal cases [33]. Frontal Horns’ Ratio is the ratio of the maximum width of the frontal horns of the lateral ventricles and inner diameter of the skull in the same line, and the Mean FHR was found to be 0.302 [34,35].Callosal Angle is also a recent imaging factor that has been suggested to be associated with hydrocephalus. But it is still need evidence of clinical application [13].

From a measurement point of view, we investigated the diagnostic effectiveness of the combined nine radiological features and morphometric parameters. Usually, a radiologist can identify hydrocephalus patients based on their imaging features without taking measurements, and our result confirms it (AUC = 0.94 ± 0.03). ER as classic diagnostic criteria also showed some advantages, in particular with identifying patients from normal patients.

In dealing with patients who have symptoms and some radiographic changes, the value of ER decreased. In contrast, the measurement of FHL and FHR become more significant. The dilation in ventricular morphology also lost significant diagnostic value. Many patients with idiopathic ventricular system enlargement have been observed in the clinic without clinical symptoms and do not require treatment. DESH, which relies on doctors’ judgment, is a powerful diagnostic tool throughout.

In clinical practice, senior patients may have complicated symptoms, some of which may miss the diagnosis of hydrocephalus. Particularly for the iNPH, which is the only reversible type of dementia. The ventricular morphology was not always reliable.

We should observe the morphology and pay more attention to recognizing DESH. For the patient with complex symptoms, the diagnostic value of FHR should be taken into consideration rather than ER value [36,37,38].

Since hydrocephalus has typical imaging manifestations, meanwhile the patient could relieve significantly after shunt surgery. To optimize the diagnosis is of great value. Although the diagnostic criteria have been widely used for many years, it is also necessary to define and optimize the weight of each indicator. Machine learning is an excellent method to provide information, especially when some diagnostic indicators become more meaningful and others become less valuable in different situations. This research method can be applied to the optimization of the diagnosis of other diseases in the future, and it can be expected to put forward the reference weight of diagnostic criteria in different situations to achieve the role of updating clinical diagnostic criteria.

This study had several limitations. First, the diagnosis of hydrocephalus relied on the attending physicians’ evaluation. At the same time, our study was a retrospective analysis, and the number of specimens was relatively low. Fortunately, we have adopted a variety of machine learning models to analyze and process the data to minimize the omission of important indicators. Multi-center prospective study with long-term follow-up will be needed to validate the model further.

## 5. Conclusions and Future Work

This study demonstrated the potential of using machine learning for the diagnosis of hydrocephalus. Both clinical parameters and important imaging features play an essential role in the diagnosis of hydrocephalus, and it is necessary to adjust the judgment indexes purposefully in identifying suspected patients. Prospective studies on this technique are underway, and more valuable findings will be available. In the future, we will combine clinical data with imaging indicators to carry out individualized treatment for patients with hydrocephalus.

## Figures and Tables

**Figure 1 brainsci-12-01484-f001:**
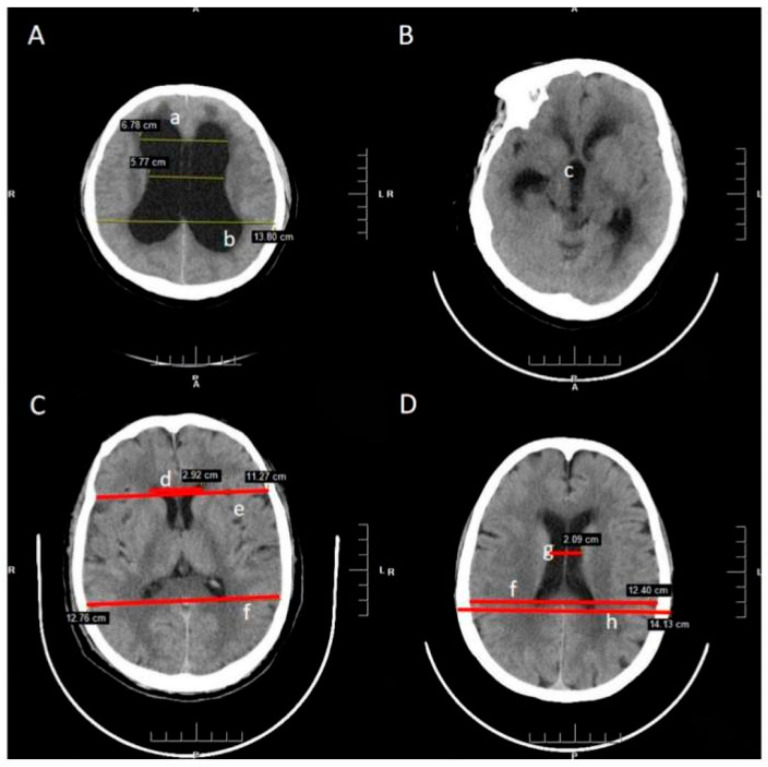
Radiological features and morphometric parameters in CT image. (**A**). The layer of enlargement ventricle; (**B**). The layer of third ventricle; (**C**). The layer of maximum width of the internal diameter (**D**). The layer of maximum transverse diameter of the skull and the narrowest width between the lateral walls (not necessarily on same level). a. Dilated temporal horns; b. Rounded posterior horns; c. Dilated third ventricle; d. Frontal Horns’ Length; e, Inner diameter of the skull in the same line as FHL; f. Maximum width of the internal diameter of the skull; g. The narrowest width between the lateral walls; h. Maximum transverse diameter of the skull.

**Figure 2 brainsci-12-01484-f002:**
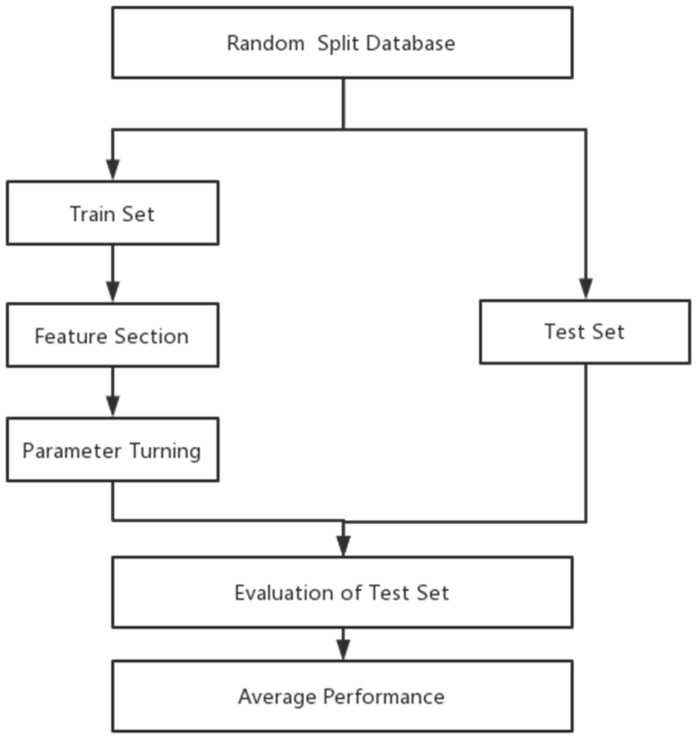
Training and evaluation procedures of the machine learning model.

**Figure 3 brainsci-12-01484-f003:**
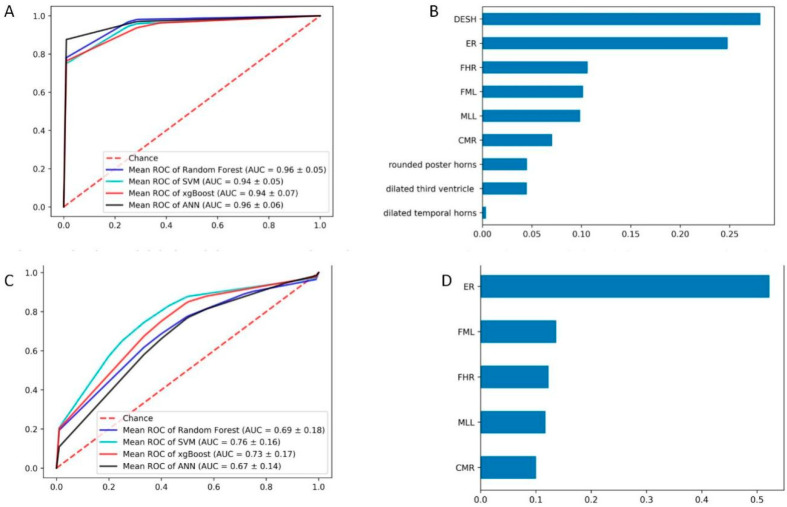
The comparison of hydrocephalus group and symptomatic group in terms of ROC and weight. (**A**) ROC curves of the four derived models with radiological features; RF, random forest, XG, XGBoost; ANN, artificial neural network; SVM, support vector machine for the patients with hydrocephalus and normal control; (**B**) The weight of all features; (**C**) ROC curves of the four derived models with morphometric parameters; (**D**) The weight of all features morphometric parameters.

**Figure 4 brainsci-12-01484-f004:**
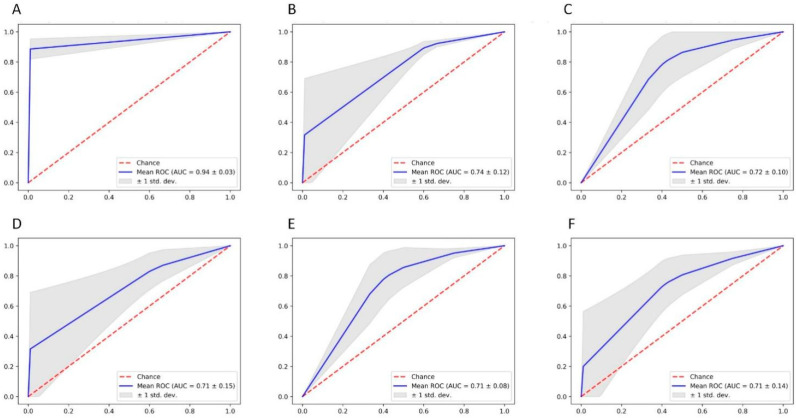
The ROC of confusion matrix diagnosis index in the comparison of hydrocephalus group and symptomatic group. (**A**) dilated temporal horns, dilated third ventricle, rounded posterior horns, and DESH; (**B**) ER + FHR; (**C**) FHR + CMR; (**D**) ER + FHR + CMR; (**E**) FHR + CMR; (**F**) ER.

**Figure 5 brainsci-12-01484-f005:**
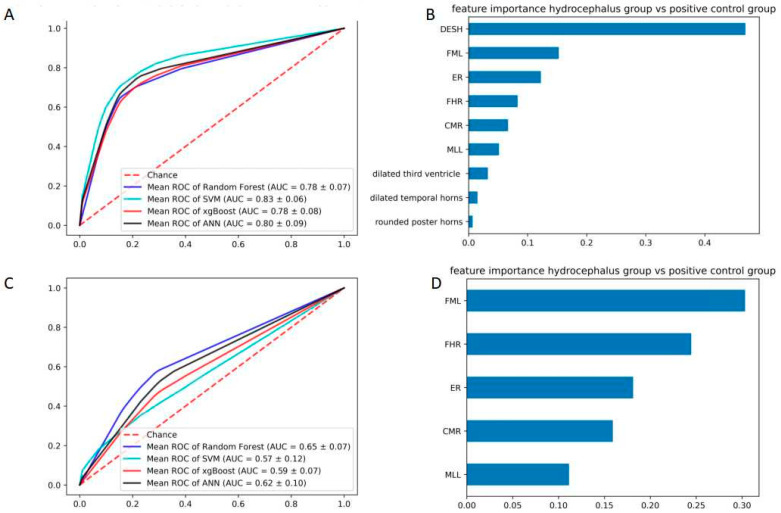
The comparison of hydrocephalus group and symptomatic group in terms of ROC and weight. (**A**) ROC curves of the four derived models with all features: RF, random forest, XG, XGBoost; ANN, artificial neural network; SVM, support vector machine for the patients with hydrocephalus and normal control; (**B**) The weight of all features; (**C**) ROC curves of the four derived models with morphometric parameters; (**D**) The weight of all features morphometric parameters.

**Table 1 brainsci-12-01484-t001:** The Radiological Features and Morphometric Parameters of CT Image.

Features	Description
DESH	Disproportionately enlarged subarachnoid space hydrocephalus
Dilated temporal horns	Whether there is temporal horns dilated
Rounded third ventricle	Whether there is third ventricle rounded
horns	Whether there is posterior horns dilated
Morphometric parameters	
MLL	The narrowest width between the lateral walls
MTD	Maximum transverse diameter of the skull
DSL	The internal diameter of the skull in the same line as MLL
BPD	Maximum width of internal diameter of the skull
DM	Inner diameter of the skull in the same line as FHL
FHL	Width of greatest span of frontal horns
ER = FHL/MTD (d/f)	The ratio of the transverse diameter of the anterior horns of the lateral ventricles to the internal diameter
CMR = MLL/BPD (g/h)	The ratio of the minimum distance between lateral walls of lateral ventricles in cella media region
FHR = FHL/DM (d/e)	The ratio of maximum width of the frontal horns of the lateral ventricles

**Table 2 brainsci-12-01484-t002:** The Baseline Statistics Enrolled Patients.

	Hydrocephalus Group (*n* = 62)	Symptomatic Group (*n* = 36)	Normal Control Group (*n* = 200)	F/χ²/Z	*p* Value
Sex (male)	39/62.90	19/52.80	11/54.00	3.010 **	0.222
age	49.87 ± 15.53	70.37 ± 11.42	52.80 ± 11.36	14.498 *	0.000
DESH	49/79.03	2/5.60	0/0	65.945 **	0.000
Dilated temporal horns	15/24.19	13/36.1	0/0	8.474 **	0.014
Dilated third ventricle	31/50.00	3/8.30	0/0	27.860 **	0.000
Rounded poster horns	18/29.03	8/22.20	0/0	8.157 **	0.017
MLL	3.71 ± 0.738	3.27 ± 0.615	2.81 ± 0.638	14.498 *	0.000
MTD	14.81 ± 1.168	14.65 ± 0.720	15.03 ± 1.039	0.855 *	0.428
DM	12.65 ± 1.15	12.34 ± 0.992	12.54 ± 0.815	1.200 *	0.305
ER	0.307 ± 0.069	0.256 ± 0.035	0.229 ± 0.023	35.274 ***	0.000
CMR	0.271 ± 0.052	0.238 ± 0.046	0.204 ± 0.047	14.835 *	0.000
FHR	0.359 ± 0.082	0.305 ± 0.041	0.275 ± 0.038	26.548 ***	0.000

* = F value; ** = χ² value; *** = Z value.

**Table 3 brainsci-12-01484-t003:** Performance Comparison of Machine Learning model in Hydrocephalus Group Compared to Normal Control Group.

All Features Model	SVM	ANN	Random Forest	xgBoost
Test Set Percision	0.928571	1.000000	0.818182	1.000000
Test Set Recall	1.000000	1.000000	0.900000	0.923077
Test Set f1	0.962963	1.000000	0.857143	0.960000
Morphometric parameters Model				
Test Set Percision	0.857143	1.000000	0.857143	0.600000
Test Set Recall	1.000000	0.866667	1.000000	0.818182
Test Set f1	0.923077	0.928571	0.923077	0.692308

**Table 4 brainsci-12-01484-t004:** Performance Comparison Confusion Matrix Diagnosis Index.

All Features Model	Radiographic Features	ER	ER + FHR	ER + CMR	FHR + CMR	ER + CMR + FHR
Test Set Percision	1.000000	0.800000	0.588235	0.923077	0.923077	0.857143
Test Set Recall	0.714286	0.666667	1.000000	0.857143	0.857143	1.000000
Test Set f1	0.833333	0.727273	0.740741	0.888889	0.888889	0.923077

**Table 5 brainsci-12-01484-t005:** Performance Comparison of Machine Learning model in Hydrocephalus Group Compared to Symptomatic Group.

All Features Model	SVM	ANN	Random Forest	xgBoost
Test Set Percision	0.900000	0.857143	0.666667	1.000000
Test Set Recall	1.000000	0.750000	0.800000	0.571429
Test Set f1	0.947368	0.800000	0.727273	0.727273
Morphometric parameters Model				
Test Set Percision	0.583333	0.333333	0.375000	0.400000
Test Set Recall	1.000000	0.222222	0.428571	0.500000
Test Set f1	0.736842	0.266667	0.400000	0.444444

**Table 6 brainsci-12-01484-t006:** The Baseline Statistics Enrolled Patients (>60).

	Hydrocephalus Group (*n* = 18)	Symptomatic Group (*n* = 29)	t/χ²	*p* Value
age	67.33 ± 4.46	73.90 ± 8.85	−3.365 *	0.002
DESH	14/77.78	1/3.45	28.239 **	0.000
Dilated temporal horns	4/22.22	10/34.48	0.798 **	0.372
Dilated third ventricle	10/55.56	3/10.34	11.346 **	0.001
Rounded poster horns	5/27.78	6/20.69	0.311 **	0.577
MLL	3.71 ± 0.738	3.27 ± 0.615	14.498 *	0.000
ER	0.29 ± 0.05	0.26 ± 0.03	2.748 *	0.011
EMR	0.27 ± 0.06	0.24 ± 0.05	2.268 *	0.028
FHR	0.35 ± 0.07	0.31 ± 0.04	2.691 *	0.010

* = t value; ** = χ² value.

## Data Availability

All data generated or analysed during this study are included in this article. Written informed consent has been obtained from the patient(s) to publish this paper.

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
