# Peer review of "Multiple Machine Learning Approaches for Morphometric Parameters in Prediction of Hydrocephalus"

_brainsci, 2022, doi:10.3390/brainsci12111484_

Round 1

Reviewer 1 Report

This study has basically a good clinically relevant idea and the control group is rather large. However, there are number of concerns.

There are two different exclusion criteria shown, these seems to be in discrepancy between the reported patients.

Line 55-78: The exclusion criteria: 1. Older than 18 years; 2. Ventricular enlargement in CT scan; 3. Consistent with symptoms related to hydrocephalus: urinary incontinence, ataxia, slow, progressive autonomous language body activity decreases, abnormal gait. Please correct carefully.

Based on the diagnostic protocol, most of the patients seems to represent normal pressure hydrocephalus (NPH), either secondary or idiopathic. The reported diagnostic criteria (ref 15) seems to be too old to be applied as such. Discussion notify iNPH but it would be essential to better classify/subgroup the patients of the current study – iNPH, sNPH, acute hydrocephalus, other?

Please give some more details of the controls. For the head CT there should always be some indication.

Figure 1 could have better resolution and should present also the other parameters presented in Table 1.

Please provide proper figure legends for all the figures.

Lines 99-101: “…artificial selection aimed at dimension reduction was used for feature extraction from thousands of variables in this analysis.” This “thousand of variables” seems a bit confusing since the next sentence refers to variable that are selected by clinicians and I suppose that this means the variable described in table 1.

Lines 131-132: “The baseline statistics of the hydrocephalus group, symptoms group, and symptomatic group are presented in table 2.” Should there be “asymptomatic” group that refers to the healthy controls? “symptoms group” and “symptomatic group” means the same in case I understand correctly.

Lines 196-197: “The diagnosis of hydrocephalus depends on clinical manifestations, imaging findings, and lumbar puncture results.” Lumbar puncture is useful especially in NPH but could be life-threatening in acute hydrocephalus especially in obstructive form. Although the obstructive cases were excluded in from this study, the lumbar puncture issue should be discussed more carefully and more specifically. The next sentence is good but this introductory clause is too undetailed.

Author Response

Thank you very much for your decision letter and advice on our manuscript entitled. We also thank the reviewers for the constructive comments and suggestions. Accordingly, we have revised the manuscript. All amendments are highlighted in red in the revised manuscript. In addition, point-by-point responses to the comments are listed below this letter.

Reviewer 2 Report

In the presented article the authors attempt to identify the association between hydrocephalus imaging and the clinical definition of such condition based on machine learning techniques, in order to clarify the role of many known imaging characteristics, nowadays widely used, in defining as certainly as possible the diagnosis of hydrocephalus.

The analysis is conducted by means of three groups: hydrocephalus group, symptomatic group and healthy control group. Such divisions have been used to compare different morphological parameters and their

statistical weight in diagnosing or excluding hydrocephalus. The inclusion and exclusion criteria for the study, as they are presented are not so definite, in particular the aspect of the age of the patients is not expressed clearly. Although the criteria to define the diagnosis of hydrocephalus are strict, only patients who responded positively to the TAP test were considered affected by the condition, while all other persons were considered in the “symptomatic group”; this aspect may have created a bias in the results, since, as the literature state, a discrete percentage of the hydrocephalus-affected population does not show an improvement after the lumbar puncture. The characteristics of the groups are altogether homogeneous, with 62 hydrocephalus-diagnosed persons, 36 symptomatic but not responsive to TAP test patients and a wide healthy population as control.

The range of imaging parameters in CT scans is wide, from Evans index to DESH and many others, offering a wide possibility of statistical analyses; undoubtedly, the inclusion of MRI imaging would have offered more precise results and even more detailed morphometric parameters.

The study then considered four different protocols of machine learning, their precision, recall and F1 values. The aspects presented by the authors show interesting results from all types of protocols with some minor differences from one to another. Furthermore, the most interesting association resulted from morphometric aspect of CT scans and the percentage of correct diagnoses obtained by the machine learning systems: Evans index, Cella media index, Front horn ratio, dilated third ventricle and DESH showed a statistically significant difference between hydrocephalus group and the other groups.

Altogether the concept of developing a useful tool based on machine learning protocols to diagnose a condition as variegated in clinical and radiological characteristics as hydrocephalus is interesting and might help in some difficult circumstances whether the symptoms are not so clear or the radiological parameters are not so definite; although by now, still no universally accepted criteria for diagnosing hydrocephalus have been recognized and many other comorbidities might also act as a confounding, especially in the elderly. Thus, any study which begins with the necessity to separate a definite hydrocephalus group from symptomatic persons without certain hydrocephalus takes into account the possibility of creating a systemic bias in the diagnostic criteria: in the case of the study only TAP test responsive persons have been considered affected by hydrocephalus.

As the authors state the fact of the study is retrospective and with a relatively low specimen might have influenced the results. An English review would also be needed

Author Response

(The authors gave the same response as above.)

Round 2

Reviewer 2 Report

Ok for publication